# FAST PARTIAL FOURIER TRANSFORM

## ABSTRACT

Given a time-series vector, how can we efficiently compute a specified part of Fourier coefficients? Fast Fourier transform (FFT) is a widely used algorithm that computes the discrete Fourier transform in many machine learning applications. Despite the pervasive use, FFT algorithms do not provide a fine-tuning option for the user to specify one's demand, that is, the output size (the number of Fourier coefficients to be computed) is algorithmically determined by the input size. Such a lack of flexibility is often followed by just discarding the unused coefficients because many applications do not require the whole spectrum of the frequency domain, resulting in an inefficiency due to the extra computation.

In this paper, we propose a fast Partial Fourier Transform (PFT), an efficient algorithm for computing only a part of Fourier coefficients. PFT approximates a part of twiddle factors (trigonometric constants) using polynomials, thereby reducing the computational complexity due to the mixture of many twiddle factors. We derive the asymptotic time complexity of PFT with respect to input and output sizes, as well as its numerical accuracy. Experimental results show that PFT outperforms the current state-of-the-art algorithms, with an order of magnitude of speedup for sufficiently small output sizes without sacrificing accuracy.

## 1 INTRODUCTION

How can we efficiently compute a specified part of Fourier coefficients of a given time-series vector? Discrete Fourier transform (DFT) is a crucial task in several application areas, including anomaly detection (Hou & Zhang (2007); Rasheed et al. (2009); Ren et al. (2019)), data center monitoring (Mueen et al. (2010)), and image processing (Shi et al. (2017)). Notably, in many such applications, it is well known that the DFT results in strong "energy-compaction" or "sparsity" in the frequency domain. That is, the Fourier coefficients of data are mostly small or equal to zero, having a much smaller support compared to the input size. Moreover, the support can often be specified in practice (e.g., a few low-frequency coefficients around the origin). These observations arouse a great interest in an efficient algorithm capable of computing only a specified part of Fourier coefficients. Accordingly, various approaches have been proposed to address the problem, which include Goertzel algorithm (Burrus & Parks (1985)), Subband DFT (Hossen et al. (1995); Shentov et al. (1995)), and Pruned FFT (Markel (1971); Skinner (1976); Nagai (1986); Sorensen & Burrus (1993); Ailon & Liberty (2009)).

In this paper, we propose a fast Partial Fourier Transform (PFT), an efficient algorithm for computing a part of Fourier coefficients. Specifically, we consider the following problem: *given a complex-valued vector $\boldsymbol{a}$ of size $N$, a non-negative integer $M$, and an integer $\mu$, estimate the Fourier coefficients of $\boldsymbol{a}$ for the interval $[\mu - M, \mu + M]$*. The resulting algorithm is of remarkably simple structure, composed of several "smaller" FFTs combined with linear pre- and post-processing steps. Consequently, PFT reduces the number of operations to $O(N + M \log M)$, which is, to the best of our knowledge, the lowest arithmetic complexity achieved so far. Besides that, most subroutines of PFT are the already highly optimized algorithms (e.g., matrix multiplication and FFT), thus the arithmetic gains are readily turned into actual run-time improvements. Furthermore, PFT does not require the input size to be a power of 2, unlike many other competitors. This is because the idea of PFT derives from a modification of the Cooley-Tukey algorithm (Cooley & Tukey, 1965), which also makes it straightforward to extend the idea to a higher dimensionality.

Through experiments, we show that PFT outperforms the state-of-the-art FFT libraries, FFTW by Frigo & Johnson (2005) and Intel Math Kernel Library (MKL) as well as Pruned FFTW, with an order of magnitude of speedup without sacrificing accuracy.

## 2 RELATED WORK

We describe various existing methods for computing partial Fourier coefficients.

**Fast Fourier transform.** One may consider just using Fast Fourier transform (FFT) and discarding the unnecessary coefficients, where FFT efficiently computes the full DFT, reducing the arithmetic cost from naïve $O(N^2)$ to $O(N \log N)$. Such an approach has two major advantages: (1) it is straightforward to implement, and (2) the method often outperforms the competitors because it directly employs FFT which has been highly optimized over decades. Therefore, we provide extensive comparisons of PFT and FFT both theoretically and through run-time evaluations. Experimental results in Section 4.2 show that PFT outperforms the FFT when the output size is small enough ($< 10\%$) compared to the input size.

**Goertzel algorithm.** Goertzel algorithm (Burrus & Parks (1985)) is one of the first methods devised for computing only a part of Fourier coefficients. The technique is essentially the same as computing the individual coefficients of DFT, thus requiring $O(MN)$ operations for $M$ coefficients of an input of size $N$. Specifically, theoretical analysis represents "the $M$ at which the Goertzel algorithm is advantageous over FFT" as $M < 2 \log N$ (Sysel & Rajmic (2012)). For example, with $N = 2^{22}$, the Goertzel algorithm becomes faster than FFT only when $M < 44$, while PFT outperforms FFT for $M < 2^{19} = 524288$ (Figure 1b). A few variants which improve the Goertzel algorithm have been proposed (e.g., Boncelet (1986)). Nevertheless, the performance gain is only by a small constant factor, thus they are still limited to rare scenarios where a very few number of coefficients are required.

**Subband DFT.** Subband DFT (Hossen et al. (1995); Shentov et al. (1995)) consists of two stages of algorithm: Hadamard transform that decomposes the input sequence into a set of smaller subsequences, and correction stage for recombination. The algorithm approximates a part of coefficients by eliminating subsequences with small energy contribution, and manages to reduce the number of operations to $O(N + M \log N)$. Apart from the arithmetic gain, however, there is a substantial issue of low accuracy with the Subband DFT. Indeed, experimental results in Hossen et al. (1995) show that the relative approximation error of the method is around $10^{-1}$ (only one significant figure) regardless of output size. Moreover, the Fourier coefficients can be evaluated to arbitrary numerical precision with PFT, which is not the case for Subband DFT. Such limitations often preclude one from considering the Subband DFT in applications that require a certain degree of accuracy.

**Pruned FFT.** FFT pruning (Markel (1971); Skinner (1976); Nagai (1986); Sorensen & Burrus (1993); Ailon & Liberty (2009)) is another technique for the efficient computation of partial Fourier coefficients. The method is a modification of the standard split-radix FFT, where the edges (operations) in a flow graph are pruned away if they do not affect the specified range of frequency domain. Besides being almost optimized (it uses FFT as a subroutine), the FFT pruning algorithm is exact and reduces the arithmetic cost to $O(N \log M)$. Thus, along with the full FFT, the pruned FFT is reasonably the most appropriate competitor of PFT. Through experiments (Section 4.2), we show that PFT consistently outperforms the pruned FFT, significantly extending the range of output sizes for which partial Fourier transform becomes practical.

Finally, we mention that there have been other approaches but with different settings. For example, Hassanieh et al. (2012a;b) and Indyk et al. (2014) propose Sparse Fourier transform, which estimates the top-$k$ (the $k$ largest in magnitude) Fourier coefficients of a given vector. The algorithm is useful especially when there is prior knowledge of the number of non-zero coefficients in frequency domain. Note that our setting does not require any prior knowledge of the given data.

**Applications of FFT.** We outline various applications of Fast Fourier transform, to which partial Fourier transform can potentially be applied. FFT has been widely used for anomaly detection (Hou & Zhang (2007); Rasheed et al. (2009); Ren et al. (2019)). Hou & Zhang (2007) and Ren et al. (2019) detect anomalous points of a given data by extracting a compact representation with FFT. Rasheed et al. (2009) use FFT to detect local spatial outliers which have similar patterns within a region but different patterns from the outside. Several works (Pagh (2013); Pham & Pagh (2013); Malik & Becker (2018)) exploit FFT for efficient operations. Pagh (2013) leverages FFT to efficiently compute a polynomial kernel used with support vector machines (SVMs). Malik & Becker (2018) propose an efficient tucker decomposition method using FFT. In addition, FFT has been used for fast training of convolutional neural networks (Mathieu et al. (2014); Rippel et al. (2015)) and an efficient recommendation model on a heterogeneous graph (Jin et al. (2020)).

## 3 PROPOSED METHOD

### 3.1 OVERVIEW

We propose PFT, an efficient algorithm for computing a specified part of Fourier coefficients. The main challenges and our approaches are as follows:

1. **How can we extract essential information for a specified output?** Considering that only a specified part of Fourier coefficients should be computed, we need to find an algorithm requiring fewer operations than the direct use of conventional FFT. This is achievable by carefully modifying the Cooley-Tukey algorithm, finding *twiddle factors* (trigonometric factors) with small oscillations, and approximating them via polynomials (Section 3.2.1).

2. **How can we reduce approximation cost?** The approach given above involves an approximating process, which would be computationally demanding. We propose using a *base exponential function*, by which all data-independent factors can be precomputed, enabling one to bypass the approximation problem during the run-time (Sections 3.2.2 and 3.3).

3. **How can we further reduce numerical computation?** We carefully reorder operations and factorize terms in order to alleviate the complexity of PFT. Such techniques separate all data-independent factors from data-dependent factors, allowing further precomputation. The arithmetic cost of the resulting algorithm is $O(N + M \log M)$, where $N$ and $M$ are input and output size descriptors, respectively (Sections 3.4 and 3.5.1).

### 3.2 APPROXIMATION OF TWIDDLE FACTORS

The key of our algorithm is to approximate a part of twiddle factors with small oscillations by using polynomial functions, reducing the computational complexity of DFT due to the mixture of many twiddle factors. Using polynomial approximation also allows one to carefully control the degree of polynomial (or the number of approximating terms), enabling fine-tuning the output range and the approximation bound of the estimation. Our first goal is to find a collection of twiddle factors with small oscillations. This can be achieved by slightly adjusting the summand of DFT and splitting the summation as in the Cooley-Tukey algorithm (Section 3.2.1). Next, using a proper base exponential function, we give an explicit form of approximation to the twiddle factors (Section 3.2.2).

### 3.2.1 TWIDDLE FACTORS WITH SMALL OSCILLATIONS

Recall that the DFT of a complex-valued vector $\boldsymbol{a}$ of size $N$ is defined as follows:

$$\hat{a}_m = \sum_{n \in [N]} a_n e^{-2\pi i m n / N}, \tag{1}$$

where $[\nu]$ denotes $\{0, 1, \cdots, \nu - 1\}$ for a positive integer $\nu$ (in this paper, we follow the convention of viewing a vector $\boldsymbol{v} = (v_0, v_1, \cdots, v_{\nu-1})$ of size $\nu$ as a finite sequence defined on $[\nu]$). Assume that $N = pq$ for two integers $p, q > 1$. The Cooley-Tukey algorithm re-expresses (1) as

$$\hat{a}_m = \sum_{k \in [p]} \sum_{l \in [q]} a_{qk+l} e^{-2\pi i m (qk+l)/N} = \sum_{k \in [p]} \sum_{l \in [q]} a_{qk+l} e^{-2\pi i m l / N} \cdot e^{-2\pi i m k / p}, \tag{2}$$

yielding two collections of twiddle factors, namely $\{e^{-2\pi i m l / N}\}_{l \in [q]}$ and $\{e^{-2\pi i m k / p}\}_{k \in [p]}$. Consider the problem of computing $\hat{a}_m$ for $-M \leq m \leq M$, where $M \leq N/2$ is a non-negative integer. In this case, note that the exponent of $e^{-2\pi i m l / N}$ ranges from $-2\pi i M (q-1)/N$ to $+2\pi i M (q-1)/N$ and that the exponent of $e^{-2\pi i m k / p}$ ranges from $-2\pi i M(p-1)/p$ to $+2\pi i M(p-1)/p$. Here $\frac{(q-1)/N}{(p-1)/p} \sim \frac{1}{p}$, meaning that the first collection contains twiddle factors with smaller oscillations compared to the second one. Typically, a function with smaller oscillation results in a better approximation via polynomials. In this sense, it is reasonable to approximate the first collection of twiddle factors in (2) with polynomial functions, thereby reducing the complexity of the computation due to the mixture of two collections of twiddle factors. Indeed, one can further reduce the complexity of approximation: we slightly adjust the summand in (2) as follows.

$$\hat{a}_m = e^{-\pi i m / p} \sum_{k \in [p]} \sum_{l \in [q]} a_{qk+l} e^{-2\pi i m (l - q/2)/N} \cdot e^{-2\pi i m k / p}. \tag{3}$$

In (3), we observe that the range of exponents of the first collection $\{e^{-2\pi i m (l-q/2)/N}\}_{l \in [q]}$ of twiddle factors is $[-\pi i M/p, +\pi i M/p]$, a contraction by a factor of around 2 when compared with $[-2\pi i M(q-1)/N, +2\pi i M(q-1)/N]$, hence the twiddle factors with even smaller oscillations. There is an extra twiddle factor $e^{-\pi i m / p}$ in (3). Note that, however, it depends on neither $k$ nor $l$, so the amount of the additional computation is relatively small.

### 3.2.2 BASE EXPONENTIAL FUNCTION

The first collection of twiddle factors in (3) consists of $q$ distinct exponential functions. One can apply approximation process for each function in the collection; however, this would be time-consuming. A more plausible approach is to 1) choose a base exponential function $e^{uix}$ for a fixed $u \in \mathbb{R}$, 2) approximate $e^{uix}$ using a polynomial, and 3) exploit a property of exponential functions: the laws of exponents. Specifically, suppose that we obtained a polynomial $\mathcal{P}(x)$ that approximates $e^{uix}$ on $|x| \leq |\xi|$, where $u, \xi \in \mathbb{R} \setminus \{0\}$. Consider another exponential function $e^{vix}$, where $v \neq 0$. Since $e^{vix} = e^{ui(vx/u)}$, the re-scaled polynomial $\mathcal{P}(vx/u)$ approximates $e^{vix}$ on $|x| \leq |u\xi/v|$. This observation indicates that once we find an approximation $\mathcal{P}$ to $e^{uix}$ on $|x| \leq |\xi|$ for properly se-lected $u$ and $\xi$, all elements belonging to $\{e^{-2\pi im(l-q/2)/N}\}_{l \in [q]}$ can be approximated by re-scaling $\mathcal{P}$. Fixing a base exponential function also enables precomputing a polynomial that approximates it, so that one can bypass the approximation problem during the run-time. We further elaborate this idea in a rigorous manner after giving a few definitions (see Definitions 3.1 and 3.2).

Let $\| \cdot \|_R$ be the uniform norm (or supremum norm) restricted to a set $R \subseteq \mathbb{R}$, that is, $\|f\|_R = \sup\{|f(x)| : x \in R\}$ and $P_\alpha$ be the set of polynomials on $\mathbb{R}$ of degree at most $\alpha$.

**Definition 3.1.** Given a non-negative integer $\alpha$ and non-zero real numbers $\xi, u$, we define a polyno-mial $\mathcal{P}_{\alpha,\xi,u}$ as the best approximation to $e^{uix}$ out of the space $P_\alpha$ under the restriction $|x| \leq |\xi|$:
$$\mathcal{P}_{\alpha,\xi,u} := \underset{P \in P_\alpha}{\arg\min} \|P(x) - e^{uix}\|_{|x| \leq |\xi|},$$
and $\mathcal{P}_{\alpha,\xi,u} = 1$ when $\xi = 0$ or $u = 0$. $\qquad\qquad\square$

Smirnov & Smirnov (1999) proved the unique existence of $\mathcal{P}_{\alpha,\xi,u}$. A few techniques called *minimax approximation algorithms* for computing the polynomial are reviewed in Fraser (1965).

**Definition 3.2.** Given a tolerance $\epsilon > 0$ and a positive integer $r \geq 1$, we define $\xi(\epsilon, r)$ to be the scope about the origin such that the exponential function $e^{\pi ix}$ can be approximated by a polynomial of degree less than $r$ with approximation bound $\epsilon$:
$$\xi(\epsilon, r) := \sup\{\xi \geq 0 : \|\mathcal{P}_{r-1,\xi,\pi}(x) - e^{\pi ix}\|_{|x| \leq \xi} \leq \epsilon\}.$$
We express the corresponding polynomial as $\mathcal{P}_{r-1,\xi(\epsilon,r),\pi}(x) = \sum_{j \in [r]} w_{\epsilon,r-1,j} \cdot x^j$. $\qquad\square$

In Definition 3.2, we choose $e^{\pi ix}$ as a base exponential function. The rationale behind is as fol-lows. First, using a minimax approximation algorithm, we precompute $\xi(\epsilon, r)$ and $\{w_{\epsilon,r-1,j}\}_{j \in [r]}$ for several tolerance $\epsilon$'s (e.g. $10^{-1}, 10^{-2}, \cdots$) and positive integer $r$'s (typically $1 \leq r \leq 25$). When $N, M, p$ and $\epsilon$ are given, we find the minimum $r$ satisfying $\xi(\epsilon, r) \geq M/p$. Then, by the pre-ceding argument, it follows that the re-scaled polynomial function $\mathcal{P}_{r-1,\xi(\epsilon,r),\pi}(-2x(l-q/2)/N)$ approximates $e^{-2\pi ix(l-q/2)/N}$ on $|x| \leq |\frac{N}{2(l-q/2)} \cdot \frac{M}{p}|$ for each $l \in [q]$ (note that if $l - q/2 = 0$, we have $|\frac{N}{2(l-q/2)} \cdot \frac{M}{p}| = \infty$). Here $|\frac{N}{2(l-q/2)} \cdot \frac{M}{p}| = |\frac{q}{2l-q} \cdot M| \geq M$ for all $l \in [q]$. Therefore, we obtain a polynomial approximation on $|m| \leq M$ for each twiddle factor in $\{e^{-2\pi im(l-q/2)/N}\}_{l \in [q]}$, namely $\{\mathcal{P}_{r-1,\xi(\epsilon,r),\pi}(-2m(l-q/2)/N)\}_{l \in [q]}$. Then, it follows from (3) that
$$\hat{a}_m \approx e^{-\pi im/p} \sum_{k \in [p]} \sum_{l \in [q]} a_{qk+l}\, \mathcal{P}_{r-1,\xi(\epsilon,r),\pi}(-2m(l-q/2)/N) \cdot e^{-2\pi imk/p}, \tag{4}$$
which gives an estimation of the coefficient $\hat{a}_m$ for $-M \leq m \leq M$.

### 3.3 ARBITRARILY CENTERED TARGET RANGES

In the previous section, we have focused on the problem of calculating $\hat{a}_m$ for $m$ belonging to $[-M, M]$. We now consider a more general case: let us use the term **target range** to indicate the range where the Fourier coefficients should be calculated, and $R_{\mu,M}$ to denote $[\mu - M, \mu + M] \cap \mathbb{Z}$, where $\mu \in \mathbb{Z}$. Note that the previously given method works only when our target range is centered at $\mu = 0$. A slight modification of the algorithm allows the target range to be arbitrarily centered. One possible approach is as follows: given a complex-valued vector $\boldsymbol{x}$ of size $N$, we define $\boldsymbol{y}$ as $y_n = x_n \cdot e^{-2\pi i\mu n/N}$. Then, the Fourier coefficients of $\boldsymbol{x}$ and $\boldsymbol{y}$ satisfy the following relationship:
$$\hat{y}_m = \sum_{n \in [N]} x_n \cdot e^{-2\pi i\mu n/N} \cdot e^{-2\pi imn/N} = \sum_{n \in [N]} x_n \cdot e^{-2\pi i(m+\mu)n/N} = \hat{x}_{m+\mu}.$$
Thus, the problem of calculating $\hat{x}_m$ for $m \in R_{\mu,M}$ is equivalent to calculating $\hat{y}_m$ for $m \in R_{0,M}$, to which our previous method can be applied. This technique, however, requires extra $N$ multiplica-

tions due to the computation of $\boldsymbol{y}$. A better approach, where one can bypass the extra process during the run-time, is to exploit the following lemma (see Appendix A.1 for the proof).

**Lemma 1.** *Given a non-negative integer $\alpha$, non-zero real numbers $\xi, u$, and a real number $\mu$, the following equality holds:*

$$e^{ui\mu} \cdot \mathcal{P}_{\alpha,\xi,u}(x-\mu) = \underset{P \in P_\alpha}{\arg\min} \|P(x) - e^{uix}\|_{|x-\mu| \le |\xi|}. \qquad \square$$

This observation implies that, in order to obtain a polynomial approximating $e^{uix}$ on $|x-\mu| \le |\xi|$, we first find a polynomial $\mathcal{P}$ approximating $e^{uix}$ on $|x| \le |\xi|$, then translate $\mathcal{P}$ by $-\mu$ and multiply it with the scalar $e^{ui\mu}$. Applying this process to the previously obtained approximation polynomials (see Section 3.2.2) yields $\{e^{-2\pi i\mu(l-q/2)/N} \cdot \mathcal{P}_{r-1,\xi(\epsilon,r),\pi}(-2(m-\mu)(l-q/2)/N)\}_{l \in [q]}$. We substitute these polynomials for the twiddle factors $\{e^{-2\pi im(l-q/2)/N}\}_{l \in [q]}$ in (3), which gives the following estimation of $\hat{a}_m$ for $m \in R_{\mu,M}$, where $k \in [p]$, $l \in [q]$, and $j \in [r]$:

$$e^{-\pi im/p} \sum_{k,l} a_{qk+l}\, e^{-2\pi i\mu(l-q/2)/N} \cdot \mathcal{P}_{r-1,\xi(\epsilon,r),\pi}(-2(m-\mu)(l-q/2)/N) \cdot e^{-2\pi imk/p}$$

$$= e^{-\pi im/p} \sum_{k,l} a_{qk+l}\, e^{-2\pi i\mu(l-q/2)/N} \sum_j w_{\epsilon,r-1,j}\, (-2(m-\mu)(l-q/2)/N)^j \cdot e^{-2\pi imk/p} \qquad (5)$$

$$= e^{-\pi im/p} \sum_j \sum_{k,l} a_{qk+l}\, e^{-2\pi i\mu(l-q/2)/N}\, w_{\epsilon,r-1,j}\, ((m-\mu)/p)^j(1-2l/q)^j \cdot e^{-2\pi imk/p}.$$

### 3.4 Efficient Summations

We have found that three main summation steps (each being over $j$, $k$ and $l$) take place when computing the partial Fourier coefficients. Note that in (5), the innermost summation $\sum_j$ is moved to the outermost position, and the term $-2(m-\mu)(l-q/2)/N$ is factorized into two independent terms, $(m-\mu)/p$ and $1-2l/q$. Interchanging the order of summations and factorizing the term result in a significant computational benefit; we elucidate what operator we should utilize for each summation and how we can save the arithmetic costs from it. As we will see, the innermost sum over $l$ corresponds to a matrix multiplication, the second sum over $k$ can be viewed as multiple DFTs, and the outermost sum over $j$ is an inner product. For the first sum, let $A = (a_{k,l}) = a_{qk+l}$ and $B = (b_{l,j}) = e^{-2\pi i\mu(l-q/2)/N}\, w_{\epsilon,r-1,j}\, (1-2l/q)^j$, so that (5) can be written as follows:

$$e^{-\pi im/p} \sum_{j \in [r]} ((m-\mu)/p)^j \sum_{k \in [p]} e^{-2\pi imk/p} \sum_{l \in [q]} a_{k,l} b_{l,j}.$$

Here, note that the matrix $B$ is data-independent (not dependent on $\boldsymbol{a}$), and thus can be precomputed. Indeed, we have already seen that $\{w_{\epsilon,r-1,j}\}_{j \in [r]}$ can be precomputed. The other factors $e^{-2\pi i\mu(l-q/2)/N}$ and $(1-2l/q)^j$ composing the elements of $B$ can also be precomputed if $(N, M, \mu, p, \epsilon)$ is known in advance. Thus, as long as the setting $(N, M, \mu, p, \epsilon)$ is unchanged, we can reuse the matrix $B$ for any input data $\boldsymbol{a}$ once the configuration phase of PFT is completed (Algorithm 1). We shall denote the multiplication $A \times B$ as $C = (c_{k,j})$:

$$e^{-\pi im/p} \sum_{j \in [r]} ((m-\mu)/p)^j \sum_{k \in [p]} c_{k,j} \cdot e^{-2\pi imk/p}. \qquad (6)$$

For each $j \in [r]$, the summation $\hat{c}_{m,j} = \sum_{k \in [p]} c_{k,j} \cdot e^{-2\pi imk/p}$ is a DFT of size $p$. We perform FFT $r$ times for this computation, which yields the following estimation of $\hat{a}_m$ for $m \in R_{\mu,M}$:

$$e^{-\pi im/p} \sum_{j \in [r]} ((m-\mu)/p)^j \cdot \hat{c}_{m,j}. \qquad (7)$$

Note that $\hat{c}_{m,j}$ is a periodic function of period $p$ with respect to $m$, so we use the coefficient at $m$ modulo $p$ if $m < 0$ or $m \ge p$. Thus, the $m^{th}$ Fourier coefficient of $\boldsymbol{a}$ can be estimated by the inner product of $((m-\mu)/p)^j$ and $\hat{c}_{m,j}$ with respect to $j$, followed by a multiplication with the extra twiddle factor $e^{-\pi im/p}$ (we also precompute $((m-\mu)/p)^j$ and $e^{-\pi im/p}$). The full computation is outlined in Algorithm 2. By these summation techniques, the arithmetic complexity is reduced to $O(N + M \log M)$ from naïve $O(MN)$, as described in Section 3.5.

### 3.5 Theoretical Analysis

We present theoretical analysis on the time complexity of PFT and its approximation bound.

---

**Algorithm 1:** Configuration phase of PFT

    **input** : Input size $N$, output descriptors $M$ and $\mu$, divisor $p$, and tolerance $\epsilon$
    **output:** Matrix $B$, divisor $p$, and numbers of rows and columns, $q$ and $r$

  **1** $q \leftarrow N/p$
  **2** $r \leftarrow \min\{r \in \mathbb{N} : \xi(\epsilon, r) \geq M/p\}$           // Use precomputed $\xi(\epsilon,r)$
  **3** **for** $(l,j) \in [q] \times [r]$ **do**
  **4**   $\Big|$   $B[l,j] \leftarrow e^{-2\pi i \mu(l-q/2)/N} \cdot w_{\epsilon,r-1,j} \cdot (1-2l/q)^j$ // Use precomputed $w_{\epsilon,r-1,j}$
  **5** **end**

---

**Algorithm 2:** Computation phase of PFT

    **input** : Vector $\boldsymbol{a}$ of size $N$, output descriptors $M$ and $\mu$, and configuration results $B, p, q, r$
    **output:** Vector $\mathcal{E}(\hat{\boldsymbol{a}})$ of estimated Fourier coefficients of $\boldsymbol{a}$ for $[\mu - M, \mu + M]$

  **1** $A[k,l] \leftarrow a_{qk+l}$ for $k \in [p]$ and $l \in [q]$
  **2** $C \leftarrow A \times B$
  **3** **for** $j \in [r]$ **do**
  **4**   $\Big|$   $\hat{C}[\cdot, j] \leftarrow \text{FFT}(C[\cdot, j])$           // FFT of $j$-th column of $C$
  **5** **end**
  **6** **for** $m \in [\mu - M, \mu + M]$ **do**
  **7**   $\Big|$   $\mathcal{E}(\hat{\boldsymbol{a}})[m] \leftarrow e^{-\pi i m/p} \sum_{j=0}^{r-1}((m-\mu)/p)^j \cdot \hat{C}[m\%p, j]$
  **8** **end**

---

### 3.5.1 TIME COMPLEXITY

We analyze the time complexity of PFT. Theorem 2 (see Appendix A.2 for the proof) shows that the time cost $T(N, M)$ of PFT, where $N$ and $M$ are input and output size descriptors, respectively, is bounded by $O(N + M \log M)$. Note that the theorem presumes that all prime factors of $N$ have a fixed upper bound. Yet, in practice, this necessity is not a big concern because one can readily control the input size with basic techniques such as zero-padding or re-sampling. Moreover, we empirically find that even if $N$ has a large prime factor, PFT still shows a promising performance (see Section 4.2). In Theorem 2, note that a positive integer is called $b$-**smooth** if none of its prime factors is greater than $b$. For example, the 2-smooth integers are equivalent to the powers of 2.

**Theorem 2.** *Fix a tolerance $\epsilon > 0$ and an integer $b \geq 2$. If $N$ is $b$-smooth, then the time complexity $T(N, M)$ of PFT has an asymptotic upper bound $O(N + M \log M)$.* $\qquad\square$

### 3.5.2 APPROXIMATION BOUND

We now give a theoretical approximation bound of the estimation via the polynomial $\mathcal{P}$. We denote the estimated Fourier coefficient of $\boldsymbol{a}$ as $\mathcal{E}(\hat{\boldsymbol{a}})$. Theorem 3 (see Appendix A.3 for the proof) states that the approximation bound over the target range is data-dependent of the total weight $\|\boldsymbol{a}\|_1$ of the original vector and the given tolerance $\epsilon$, where $\|\cdot\|_1$ denotes the $\ell_1$ norm.

**Theorem 3.** *Given a tolerance $\epsilon > 0$, the following inequality holds for PFT:*
$$\|\hat{\boldsymbol{a}} - \mathcal{E}(\hat{\boldsymbol{a}})\|_{R_{\mu,M}} \leq \|\boldsymbol{a}\|_1 \cdot \epsilon. \qquad\square$$

## 4 EXPERIMENTS

Through experiments, the following questions should be answered:

- **Q1. Run-time cost (Section 4.2).** How quickly does PFT compute a part of Fourier coefficients compared to other competitors without sacrificing accuracy?
- **Q2. Effect of hyper-parameter $p$ (Section 4.3).** How do the different choices of divisor $p$ of input size $N$ affect the overall performance of PFT?
- **Q3. Effect of different precision (Section 4.4).** How do the different precision settings affect the run-time of PFT?
- **Q4. Anomaly detection (Section 4.5).** How well does PFT work for a practical application employing FFT (anomaly detection)?

### 4.1 EXPERIMENTAL SETUP

**Machine.** A machine with Intel Core i7-6700HQ @ 2.60GHz and 8GB of RAM is used.

**Datasets.** We use both synthetic and real-world datasets listed in Table 1.

Table 1: Detailed information of datasets.

| Dataset | Type | Size | Description |
|---|---|---|---|
| $\{S_n\}_{n=12}^{22}$ | Synthetic | $2^n$ | Vectors of random real numbers between 0 and 1 |
| Urban Sound[1] | Real-world | 32000 | Various sound recordings in urban environment |
| Air Condition[2] | Real-world | 19735 | Time-series vectors of air condition information |

**Competitors.** We compare PFT with two state-of-the-art FFT algorithms, FFTW and MKL, as well as Pruned FFTW. All of them are implemented in C++.

1. **FFTW**: FFTW[3] is one of the fastest public implementation for FFT, offering a hardware-specific optimization. We use the optimized version of FFTW 3.3.5, and do not include the pre-processing for the optimization as the run-time cost.
2. **MKL**: Intel Math Kernel Library[4] (MKL) is a library of optimized math routines including FFT, and often shows a better run time result than the FFTW. All the experiments are conducted with an Intel processor for the best performance.
3. **pFFT**: pFFT[5] is a pruned version of FFTW designed for fast computation of a subset of the outputs. The algorithm uses the optimized FFTW as a subroutine.
4. **PFT (proposed)**: we use MKL BLAS routines for the matrix multiplication, MKL DFTI functions for the batch FFT computation, and Intel Integrated Performance Primitives (IPP) library for the post-processing steps such as inner product and element-wise multiplication.

**Measure.** In all experiments, we use single-precision floating-point format, and the parameters $p$ and $\epsilon$ are chosen so that the relative $\ell_2$ error is strictly less than $10^{-6}$, which ensures that the overall estimated coefficients have at least 6 significant figures. Explicitly,

$$\text{Relative } \ell_2 \text{ Error} = \sqrt{\frac{\sum_{m \in \mathcal{R}} |\hat{a}_m - \mathcal{E}(\hat{a})_m|^2}{\sum_{m \in \mathcal{R}} |\hat{a}_m|^2}} < 10^{-6},$$

where $\hat{a}$ is the actual coefficient, $\mathcal{E}(\hat{a})$ is the estimation of $\hat{a}$, and $\mathcal{R}$ is the target range. Section 4.4 an exception, where we investigate different settings, varying the precision to $10^{-4}$ or $10^{-2}$.

## 4.2 RUN-TIME COST

**Run time vs. input size.** We fix the target range to $R_{0,2^9}$ and evaluate the run time of PFT vs. input sizes $N : 2^{12}, 2^{13}, \cdots, 2^{22}$. Figure 1a shows how the four competitive algorithms scale with varying input size, wherein PFT outperforms the others if the output size is sufficiently smaller ($< 10\%$) than the input size. Consequently, PFT achieves up to $19\times$ speedup compared to its competitors. Due to the overhead of the $O(N)$ pre- and $O(M)$ post-processing steps, PFT runs slower than FFT when $M$ is close to $N$ so the time complexity tends to $O(N + N \log N)$.

**Run time vs. output size.** We fix the input size to $N = 2^{22}$ and evaluate the run time of PFT vs. target ranges $R_{0,2^9}, R_{0,2^{10}}, \cdots, R_{0,2^{18}}$. The result is illustrated as a run time vs. output size plot (recall that $|R_{0,M}| \simeq 2M$) in Figure 1b. Note that the run times of FFTW and MKL do not benefit from the information of output size. We also observe that the pruned FFT (pFFT) shows only a modest improvement compared to the full FFTs, while PFT significantly extends the range of output sizes for which partial Fourier transform becomes practical.

**Real-world data.** When it comes to real-world data, it is not generally the case that the size of an input vector is a power of 2. Notably, PFT still shows a promising performance regardless of the fact that the input size is not a power of 2 or even it has a large prime factor: a strong indication that our proposed technique is robust for many different applications in real-world. *Urban Sound* dataset contains various sound recording vectors of size $N = 32000 = 2^8 \times 5^3$. We evaluate the run time of PFT vs. output size ranging from 100 to 6400. Figure 2a illustrates the result, wherein PFT outperforms the competitors if the output size is small enough compared to the input size. On the other hand, *Air Condition* dataset is composed of time-series vectors of size $N = 19735 = 5 \times 3947$. Note that $N$ has only two non-trivial divisors, namely 5 and 3947, forcing one to choose $p = 3947$ in any practical settings; if we choose $p = 5$, the ratio $M/p$ often turns out to be *too large*, which

---

[1]https://urbansounddataset.weebly.com/urbansound8k.html

[2]https://archive.ics.uci.edu/ml/datasets/Appliances+energy+prediction

[3]http://www.fftw.org/index.html

[4]http://software.intel.com/mkl

[5]http://www.fftw.org/pruned.html

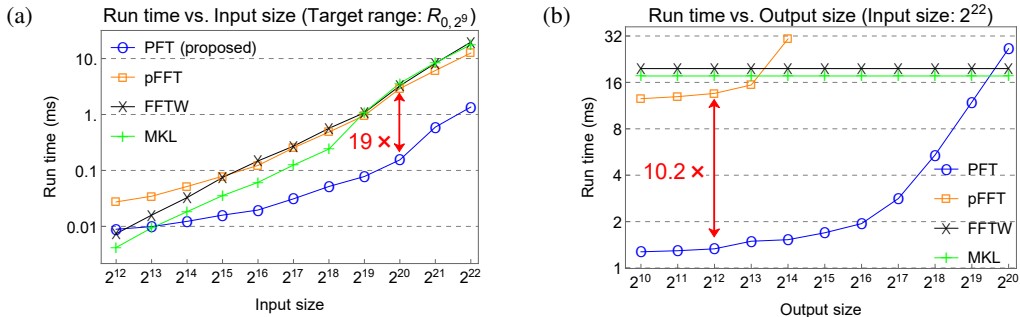

Figure 1: **(a) Run time vs. input size** for target range $R_{0,2^9}$ with $\{S_n\}_{n=12}^{22}$ datasets, and **(b) run time vs. output size** for $S_{22}$. PFT runs faster than the competitors if the output size is small enough $(< 10\%)$ compared to the input size. Note that PFT consistently outperforms the pruned FFT.

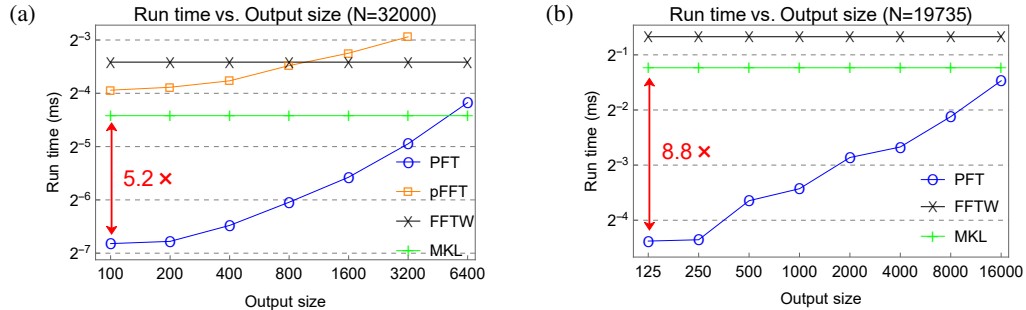

Figure 2: Run time vs. output size results for **(a) Urban Sound** dataset and **(b) Air Condition** dataset. PFT outperforms the competitors regardless of the fact that the input size is not a power of 2 $(N = 2^8 \times 5^3)$ or even it has a large prime factor $(N = 5 \times 3947)$.

results in a poor performance (see Section 4.3 for more discussion of the optimal choice of $p$). The run time of PFT vs. output size ranging from 125 to 16000 is evaluated in Figure 2b (pFFT is not included in the figure since it consistently runs slower than FFTW). It is noteworthy that PFT still outperforms its competitors even in such pathological examples, implying the robustness of our algorithm for various real-world situations.

### 4.3 EFFECT OF HYPER-PARAMETER $p$

To investigate the effect of different choices of $p$, we fix $N = 2^{22}$ and vary the ratio $M/p$ from $1/32$ to $4$ for target ranges $R_{0,2^9}, R_{0,2^{10}}, \cdots, R_{0,2^{18}}$. Table 2 shows the resulting run times, where the bold highlights the best choice of $M/p$ for each $M$, and the missing entries are due to worse performance than FFT. One crucial observation is as follows: with the increase of output size, the best choice of $M/p$ also increases or, equivalently, the optimal value of $p$ tends to remain stable. Intuitively, this is the consequence of "balancing" the three summation steps (Section 3.4): when $M \ll N$, the most computationally expensive operation is the matrix multiplication with $O(rN)$ time cost, and thus, $M/p$ should be small so that the $r$ decreases, despite a sacrifice in the batch FFT step requiring $O(rp \log p)$ operations (Appendix A.2). As the $M$ becomes larger, however, more concern is needed regarding the batch FFT and post-processing steps, so the parameter $p$ should not change rapidly. This observation indicates the possibility that the optimal value of $p$ can be algorithmically auto-selected given a setting $(N, M, \mu, \epsilon)$, which we leave as a future work.

### 4.4 EFFECT OF DIFFERENT PRECISION

We investigate the trade-off between accuracy and running time of PFT. To do this, we fix $N = 2^{22}$ and change the precision goal from $10^{-6}$ to $10^{-4}$ or $10^{-2}$ for target ranges $R_{0,2^9}, R_{0,2^{10}}, \cdots, R_{0,2^{18}}$. Table 3 shows the results, where the parenthesized number is the ratio of run times of each setting to $10^{-6}$. Note that the run time of PFT is reduced by up to 17% or 27% when the precision goal is set to $10^{-4}$ or $10^{-2}$, respectively. This observation indicates that one may readily benefit from the trade-off, especially when the fast evaluation is of utmost importance albeit with a slight sacrifice in accuracy.

Table 2: Average run time (ms) of PFT for $N = 2^{22}$ with different settings of $M/p$ and $M$.

| $M/p$ | $M$ | | | | | | | | | |
|---|---|---|---|---|---|---|---|---|---|---|
| | $2^9$ | $2^{10}$ | $2^{11}$ | $2^{12}$ | $2^{13}$ | $2^{14}$ | $2^{15}$ | $2^{16}$ | $2^{17}$ | $2^{18}$ |
| 1/32 | **1.273** | **1.394** | 1.634 | 2.303 | 5.659 | 14.121 | - | - | - | - |
| 1/8 | 2.674 | 1.608 | **1.332** | **1.491** | 1.860 | 3.020 | 7.711 | - | - | - |
| 1/2 | 2.627 | 3.738 | 2.717 | 1.678 | **1.526** | 1.881 | 2.707 | 5.740 | 14.715 | - |
| 1 | 2.677 | 2.685 | 3.805 | 2.808 | 1.687 | **1.692** | 2.164 | 3.530 | 7.749 | - |
| 2 | 4.005 | 2.723 | 2.731 | 3.533 | 2.878 | 1.949 | **1.940** | **2.821** | 5.556 | 12.534 |
| 4 | 4.090 | 4.295 | 2.986 | 2.983 | 4.108 | 3.275 | 2.365 | 2.929 | **5.411** | **11.924** |

Table 3: Average run time (ms) of PFT for $N = 2^{22}$ with different precision settings.

| Precision | $M$ | | | | | | | | | |
|---|---|---|---|---|---|---|---|---|---|---|
| | $2^9$ | $2^{10}$ | $2^{11}$ | $2^{12}$ | $2^{13}$ | $2^{14}$ | $2^{15}$ | $2^{16}$ | $2^{17}$ | $2^{18}$ |
| $10^{-6}$ | 1.273 | 1.295 | 1.332 | 1.491 | 1.526 | 1.692 | 1.940 | 2.821 | 5.411 | 11.924 |
| $10^{-4}$ | 1.249 | 1.278 | 1.293 | 1.329 | 1.400 | 1.607 | 1.872 | 2.469 | 4.590 | 9.927 |
| | (.98) | (.99) | (.97) | (.89) | (.92) | (.95) | (.96) | (.88) | (.85) | (.83) |
| $10^{-2}$ | 1.238 | 1.244 | 1.251 | 1.277 | 1.343 | 1.512 | 1.740 | 2.297 | 4.058 | 8.733 |
| | (.97) | (.96) | (.94) | (.86) | (.88) | (.89) | (.90) | (.81) | (.75) | (.73) |

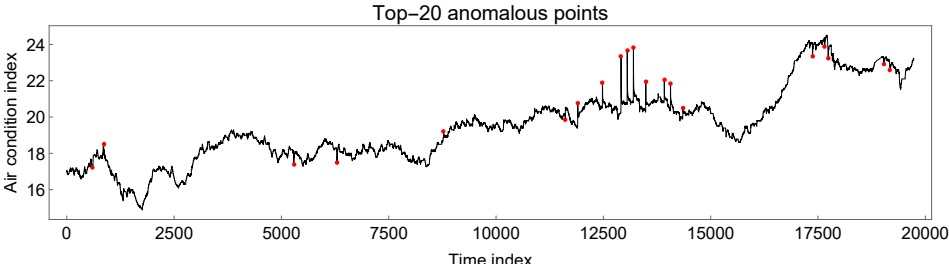

Figure 3: Top-20 anomalous points detected in Air Condition time-series data, where each red dot denotes a detected anomaly position. Note that replacing FFT with PFT does not change the result of the detection, still reducing the overall time complexity.

### 4.5 ANOMALY DETECTION

We demonstrate an example of how PFT is applied to practical applications. Here is one simple but fundamental principle: *replace the "perform FFT and discard unused coefficients" procedure with "just perform PFT"*. Considering the anomaly detection method proposed in Rasheed et al. (2009), where one first performs FFT and then inverse FFT with only a few low-frequency coefficients to obtain an estimated fitted curve, we can directly apply the principle to the method. To verify this experimentally, we use a time-series vector from Air Condition dataset, and set the target range as $R_{0,125}$ ($\simeq 250$ low-frequency coefficients). Note that, in this setting, PFT results in around $8\times$ speedup compared to the conventional FFT (see Figure 2b). The top-20 anomalous points detected from the data are presented in Figure 3. In particular, we found that replacing FFT with PFT does not change the result of top-20 anomaly detection, with all its computational benefits.

## 5 CONCLUSIONS

In this paper, we propose PFT (fast Partial Fourier Transform), an efficient algorithm for computing a part of Fourier coefficients. PFT approximates some of twiddle factors with relatively small oscillations using polynomials, reducing the computational complexity of DFT due to the mixture of many twiddle factors. Experimental results show that PFT outperforms the state-of-the-art FFTs as well as pruned FFT, with an order of magnitude of speedup without accuracy loss, significantly extending the range of applications where partial Fourier transform becomes practical. Future works include optimizing the implementation of PFT; for example, the optimal divisor $p$ of input size $N$ might can be algorithmically auto-selected. We also believe that hardware-specific optimizations would further increase the performance of PFT.

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

# A PROOFS

## A.1 PROOF OF LEMMA 1

*Proof.* Let $Q = \arg\min_{P \in P_\alpha} \|P(x) - e^{uix}\|_{|x-\mu| \leq |\xi|}$. We first observe the following equation:

$$
\begin{aligned}
Q &= \arg\min_{P \in P_\alpha} \|P(x) - e^{uix}\|_{|x-\mu| \leq |\xi|} \\
&= \arg\min_{P \in P_\alpha} \|P(x + \mu) - e^{ui(x+\mu)}\|_{|x| \leq |\xi|} \\
&= \arg\min_{P \in P_\alpha} \|e^{-ui\mu} \cdot P(x + \mu) - e^{uix}\|_{|x| \leq |\xi|},
\end{aligned}
$$

where the third equality holds since $|e^{-ui\mu}| = 1$. Recall that the polynomial $\mathcal{P}_{\alpha,\xi,u}$ is defined by $\arg\min_{P \in P_\alpha} \|P(x) - e^{uix}\|_{|x| \leq |\xi|}$. If $Q(x) \in P_\alpha$, it is clear that $e^{-ui\mu} \cdot Q(x + \mu) \in P_\alpha$ because translation and non-zero scalar multiplication on a polynomial do not change its degree. Therefore, by the uniqueness of the best approximation (Smirnov & Smirnov, 1999), we have

$$
e^{-ui\mu} \cdot Q(x + \mu) = \mathcal{P}_{\alpha,\xi,u}(x),
$$

which yields $Q(x) = e^{ui\mu} \cdot \mathcal{P}_{\alpha,\xi,u}(x - \mu)$, and hence the proof. $\qquad\square$

## A.2 PROOF OF THEOREM 2

*Proof.* We first claim that the following statement holds: *let $b \geq 2$; if $N$ is $b$-smooth and $M \leq N$ is a positive integer, then there exists a positive divisor $p$ of $N$ satisfying $M/\sqrt{b} \leq p < \sqrt{b}M$.* Indeed, suppose that none of $N$'s divisors belongs to $[M/\sqrt{b}, \sqrt{b}M)$. Let $1 = p_1 < p_2 < \cdots < p_d = N$ be the enumeration of all positive divisors of $N$ in increasing order. It is clear that $p_1 < \sqrt{b}M$ and $M/\sqrt{b} < p_d$ since $b \geq 2$ and $1 \leq M \leq N$. Then, there exists an $i \in \{1, 2, \cdots, d-1\}$ so that $p_i < M/\sqrt{b}$ and $p_{i+1} \geq \sqrt{b}M$. Since $N$ is $b$-smooth and $p_i < N$, at least one of $2p_i, 3p_i, \cdots, bp_i$ must be a divisor of $N$. However, this is a contradiction because we have $p_{i+1}/p_i > (\sqrt{b}M)(M/\sqrt{b})^{-1} = b$, so none of $2p_i, 3p_i, \cdots, bp_i$ can be a divisor of $N$, which completes the proof.

Exploiting the above property, we manage to reduce the time complexity of PFT to a functional form dependent of only $N$ and $M$. We follow the convention in counting FFT operations, assuming that all data-independent elements such as configuration results $B, p, q, r$ and twiddle factors are precomputed, and thus not included in the run-time cost. We begin with the construction of the matrix $A$. For this, we merely interpret $a$ as an array representation for $A$ of size $p \times q = N$ (line 1 in Algorithm 2). Also, recall that the matrix $B$ can be precomputed as described in Section 3.4. For the two matrices $A$ of size $p \times q$ and $B$ of size $q \times r$, standard matrix multiplication algorithm has running time of $O(pqr) = O(r \cdot N)$ (line 2 in Algorithm 2). Next, the expression (6) contains $r$ DFTs of size $p$, namely $\hat{c}_{m,j} = \sum_{k \in [p]} c_{k,j} \cdot e^{-2\pi imk/p}$ for each $j \in [r]$. We use FFT $r$ times for the computation, then it is easy to see that the time cost is given by $O(r \cdot p \log p)$ (lines 3-5 in Algorithm 2). Finally, there are $2M + 1$ coefficients to be calculated in (7), each requiring $O(r)$ operations, giving an upper bound $O(r \cdot M)$ for the running time (lines 6-8 in Algorithm 2). Combining the three upper bounds, we formally express the time complexity $T(N, M)$ of PFT as

$$
T(N, M) = O(r \cdot (N + p \log p + M)).
$$

Note that $r$ is only dependent of $\epsilon$ and $M/p$ by its definition (Algorithm 1). Therefore, when $\epsilon$ is fixed, $T(N, M)$ is dependent of the choice of $p$. By the preceding argument, we can always find a divisor $p$ of $N$ such that $M/\sqrt{b} \leq p < \sqrt{b}M$, implying that $M/p$ is tightly bounded, and thus, so is $r$. It follows that $p = \Theta(M)$ and $r = \Theta(1)$, which leads to the following asymptotic upper bound for $T(N, M)$ with respect to $N$ and $M$:

$$
T(N, M) = O(N + M \log M),
$$

hence the proof. $\qquad\square$

### A.3 PROOF OF THEOREM 3

*Proof.* Let $v = -2(l - q/2)/N$. By the estimation in (5), the following holds:

$$\|\hat{\boldsymbol{a}} - \mathcal{E}(\hat{\boldsymbol{a}})\|_{R_{\mu,M}} = \|\sum_{k,l} a_{qk+l}\big(e^{\pi i v m} - e^{\pi i v \mu} \cdot \mathcal{P}_{r-1,\xi(\epsilon,r),\pi}(v(m-\mu))\big)e^{-2\pi i m k/p}e^{-\pi i m/p}\|_{m \in R_{\mu,M}}$$

$$\leq \sum_{k,l} \|a_{qk+l}\big(e^{\pi i v m} - e^{\pi i v \mu} \cdot \mathcal{P}_{r-1,\xi(\epsilon,r),\pi}(v(m-\mu))\big)e^{-2\pi i m k/p}e^{-\pi i m/p}\|_{m \in R_{\mu,M}}$$

$$= \sum_{k,l} |a_{qk+l}| \cdot \|e^{\pi i v(m-\mu)} - \mathcal{P}_{r-1,\xi(\epsilon,r),\pi}(v(m-\mu))\|_{m \in R_{\mu,M}},$$

since we have $|e^{-2\pi i m k/p}| = |e^{-\pi i m/p}| = |e^{-\pi i v \mu}| = 1$. If $l$ ranges from 0 to $q - 1$, then $|v| \leq 2(q/2)/N = 1/p$, and thus, $M|v| \leq M/p \leq \xi(\epsilon, r)$. We extend the domain of the function $e^{\pi i v(m-\mu)} - \mathcal{P}_{r-1,\xi(\epsilon,r),\pi}(v(m-\mu))$ from $m \in R_{\mu,M} = [\mu - M, \mu + M] \cap \mathbb{Z}$ to $x \in [\mu - M, \mu + M]$ (note that extending domain never decreases the uniform norm), and replace $v(x - \mu)$ with $x'$, from which it follows that

$$\|\hat{\boldsymbol{a}} - \mathcal{E}(\hat{\boldsymbol{a}})\|_{R_{\mu,M}} \leq \sum_{k,l} |a_{qk+l}| \cdot \|e^{\pi i v(x-\mu)} - \mathcal{P}_{r-1,\xi(\epsilon,r),\pi}(v(x-\mu))\|_{|x-\mu| \leq M}$$

$$= \sum_{k,l} |a_{qk+l}| \cdot \|e^{\pi i x'} - \mathcal{P}_{r-1,\xi(\epsilon,r),\pi}(x')\|_{|x'| \leq M|v|}$$

$$\leq \sum_{k,l} |a_{qk+l}| \cdot \|e^{\pi i x'} - \mathcal{P}_{r-1,\xi(\epsilon,r),\pi}(x')\|_{|x'| \leq \xi(\epsilon,r)}$$

$$\leq \sum_{k,l} |a_{qk+l}| \cdot \epsilon$$

$$= \|\boldsymbol{a}\|_1 \cdot \epsilon,$$

where the second inequality holds since $M|v| \leq \xi(\epsilon, r)$, hence the desired result. $\qquad\square$

