# OpenReview forum: "Fast Partial Fourier Transform"
_ICLR.cc/2021/Conference — Reject_

### Official Review · AnonReviewer4 · 2020-10-28
**Neat algorithm, but comparison to prior work highly incomplete**

**Rating:** 5
**Confidence:** 4

**Review:**

The paper presents a fast approximate algorithm for partial discrete Fourier transform. Given the input signal $a_0...a_N$ in the time domain the algorithm approximately computes the first $O(M)$ frequencies. The running time is $O(r (N + M \log M))$, where $r$ a parameter that controls the accuracy of the output. The algorithm itself proceeds by applying polynomial approximation to the twiddle coefficients in FFT, which makes it possible to reduce the running time.

Pros:
+ Well-motivated problem: in many applications, most of the signal energy is present in the first few coefficients  and this algorithm computes them faster (albeit approximately)
+ To the best of my knowledge the algorithm is new, although cannot tell for sure given a large body of prior work on this topic (see below)
+ Experimental evaluation shows one order of magnitude improvement in the running time, while keeping the relative error to $10^{-6}$.

Cons:
- The papers omits a significant body of prior work on computing the first few DFT coefficients faster. For example:
  * Pruned FFT: the algorithm evaluates the first M coefficients in O(N log M) time; here, the results are *exact*. See e.g., http://www.fftw.org/pruned.html
  * Subband FFT: this method is approximate; see e.g., these papers:
Subband DFT—part II: accuracy, complexity and applications, AN Hossen, U Heute, OV Shentov, SK Mitra - Signal Processing, 1995 - Elsevier
 Subband DFT—Part I: Definition, interpretation and extensions, OV Shentov, SK Mitra, U Heute, AN Hossen - Signal Processing, 1995 - Elsevier
 Subband DFT-interpretation, accuracy, and computational complexity, OV Shentov, AN Hossen, SK Mitra… - Conference Record of the …, 1991 - computer.org

  There are also other methods cited in those articles.

- I could not find any analysis of the trade-off between the accuracy and the values of parameters r and p. As a result, it is not clear what the runtime dependence on those parameters is.

The above two points make it impossible to compare the proposed algorithm relative to what was known before, either in theory or in practice.

Additional comments:
- One can  assume (essentially) without loss of generality that the target frequencies lie in the interval centered at zero (i.e., that $\mu$ is zero), as otherwise one can use Shift Theorem to shift the frequencies by multiplying each coefficient of the time signal by a phase term. This should simplify the presentation.


Overall: the paper has interesting (and potentially very useful) ideas. However, it needs a very substantial rewrite, as per the comments above.

---

> ### Author Response · Authors · 2020-11-23
> **Response to AnonReviewer4**
>
> We would like to thank you for high quality review and constructive comments.
>
> **Comparison with existing methods**
>
> As the reviewer mentioned, we added the discussion of various existing methods including Pruned FFT and Subband DFT in Section 2 of the revised paper. Additionally, we added the Goertzel algorithm which is one of the first methods devised for evaluating a part of Fourier coefficients in the same section. In the discussion, we showed that PFT outperforms the Goertzel algorithm, which demonstrates the benefits of our proposed method. Moreover, we added the experimental comparison between PFT and Pruned FFT in Section 4.2 of the revised paper. Figures 1(a), 1(b), and 2(a) of the revised paper show that PFT outperforms Pruned FFT, significantly extending the range of output sizes for which partial Fourier transform becomes practical.
>
> **Exact or approximate?**
>
> The reviewer pointed out that Pruned FFT is exact, while our method is approximate. We claim that, in terms of practical use, the approximate algorithm is the more advantageous one as long as it provides the option to set an arbitrary approximation error bound. Indeed, the Fourier coefficients can be evaluated to arbitrary numerical precision with PFT by changing hyper-parameters, providing more versatility than the exact algorithm. Also, in the experiments in Section 4.2 of the revised paper, PFT was set to the same precision (<$10^{-6}$) as Pruned FFT, and showed up to 19 times faster running time than Pruned FFT. Therefore, one can reasonably argue that it is not a problem that PFT is approximate.
>
> **Trade-off between accuracy and runtime**
>
> As the reviewer mentioned, we added the experimental results concerning the trade-off between accuracy and runtime of PFT in Section 4.4 of the revised paper. The results show that the runtime of PFT is reduced by up to 17% and 27% as the accuracy goal changes from $10^{-6}$ to $10^{-4}$ and $10^{-2}$, respectively.
>
> **The reason why PFT does not use Shift Theorem**
>
> The reviewer also pointed out in the additional comments that one can assume essentially without loss of generality that the target range is centered at zero with the help of Shift Theorem, which is true. However, as we had already mentioned in the lines 5-12 of Section 3.3 of the original paper (also lines 5-12 of Section 3.3 of the revised paper), the technique requires extra $N$ (the input size) complex multiplications. By using our proposed method, one can bypass the extra computations during the run-time.

---

### Official Review · AnonReviewer3 · 2020-10-30
**An algorithm for quickly approximating partial Fourier transforms**

**Rating:** 5
**Confidence:** 3

**Review:**

The paper suggests a method for quickly computing a "partial Fourier transform", which basically means that we want only a small range of output frequencies.  The main technique is an approximation of so called "twiddle functions" (which are basically trigonometric functions, or, exponents of complex units if viewed in the complex plane) using polynomials.   The resulting algorithms run in time O(N + M log M) where M is the size of the required frequency range, and N is the input.  This should be compared with Cooley and Tukey's FFT, which is O(N log N).  In fact, the main idea in the paper uses Cooley and Tukey's decomposition of the expression for Fourier transform.

The first thing I must note is, that the authors seem not to be aware of existing literature on partial Fourier transforms.  For example, Ailon and Liberty show in their paper [1] that whenever you are only interested in M Fourier frequencies then you can do exact FFT in time O(N log M).  This is done by pruning Cooley Tukey's FFT.  It is not as good as O(N + M log M), but it is EXACT, and it doesn't require the M frequencies to be in a single consecutive range.  (In fact, [1] does this for the Hadamard transform, but I am quite certain that it works for the DFT as well because the argument depends on the computational graph only, which is identical for both cases).

Additionally, Indyk et al have written many papers on fast computation of Fourier when you know a priori that only M frequencies are nonzeros (and the rest are 0).  This setting is different from the one in this paper, but it is definitely relevant.

Finally, I am inclined to say that the subject of this paper is only loosely related to the scope if ICLR.  I mean, computing Fourier coefficients is definitely a way to get a useful representation of the data, but I don't know if ICLR is the correct venue for this kind of paper.

Otherwise, the paper is well written, and I assume it is correct.

[1] Ailon, Liberty, "Fast Dimension Reduction Using Rademacher Series on Dual BCH Codes. " (Discrete and Computational Geometry, 2009)

---

> ### Author Response · Authors · 2020-11-23
> **Response to AnonReviewer3**
>
> We would like to thank you for high quality review and constructive comments.
>
> **Comparison with existing works**
>
> As the reviewer mentioned, we added the discussion of various existing works including Goertzel algorithm, Subband DFT, Pruned FFT, and Sparse Fourier transform in Section 2 of the revised paper. Moreover, we also added the experimental comparison between PFT and Pruned FFT in Section 4.2 of the revised paper. Figures 1(a), 1(b), and 2(a) of the revised paper show that PFT outperforms Pruned FFT, significantly extending the range of output sizes for which partial Fourier transform becomes practical.
>
> **Exact or approximate?**
>
> The reviewer pointed out that Pruned FFT is exact, while our method is approximate. We claim that, in terms of practical use, the approximate algorithm is the more advantageous one as long as it provides the option to set an arbitrary approximation error bound. Indeed, the Fourier coefficients can be evaluated to arbitrary numerical precision with PFT by changing hyper-parameters, providing more versatility than the exact algorithm. Also, in the experiments in Section 4.2 of the revised paper, PFT was set to the same precision ($<10^{-6}$) as Pruned FFT, and showed up to 19 times faster running time than Pruned FFT. Therefore, one can reasonably argue that it is not a problem that PFT is approximate.
>
> **Consecutiveness constraint**
>
> The reviewer also pointed out that Pruned FFT does not require the output to be a single consecutive range, while PFT does. In most practical use cases, however, it is a consecutive low-frequency range where we are interested in. Besides, PFT can be used multiple times for distinct target ranges to compute the coefficients in a non-consecutive range. Thus, we argue that the reviewer’s point is not of much concern in terms of practical use.
>
> **Scope of the paper**
>
> Many papers presented in learning conferences, like ICML, CVPR, and KDD, exploit Fast Fourier transform (FFT). Our proposed approach, PFT, allows one to improve the performance (e.g., efficiency) of those existing learning approaches. Moreover, PFT will continue to support various research projects that suffer from the inefficiency of FFT. Therefore, we are confident that many ICLR audiences will be interested in PFT. The following references utilize FFT:
> - [1] Frank et al. Leveraging Frequency Analysis for Deep Fake Image Recognition, ICML 2020, https://arxiv.org/pdf/2003.08685.pdf
> - [2] Li et al. FALCON: A Fourier Transform Based Approach for Fast and Secure Convolutional Neural Network Predictions, CVPR 2020, https://openaccess.thecvf.com/content_CVPR_2020/papers/Li_FALCON_A_Fourier_Transform_Based_Approach_for_Fast_and_Secure_CVPR_2020_paper.pdf
> - [3] Yanchao Yang and Stefano Soatto. FDA: Fourier Domain Adaptation for Semantic Segmentation, CVPR 2020, https://openaccess.thecvf.com/content_CVPR_2020/papers/Yang_FDA_Fourier_Domain_Adaptation_for_Semantic_Segmentation_CVPR_2020_paper.pdf
> - [4] Chen et al. Compressing Convolutional Neural Networks in the Frequency Domain, KDD 2016, https://www.kdd.org/kdd2016/papers/files/rpp0534-chenA.pdf

---

### Official Review · AnonReviewer1 · 2020-10-31
**Interesting paper, but out of scope**

**Rating:** 6
**Confidence:** 5

**Review:**

Overall this is an interesting paper. The paper proposes a few approximations in the computation of the DFT that enable to compute the DTFT of a signal at uniform grid of 2M+1 frequency points around a certain midpoint $\mu$. While this is easy and straightforward using the direct computation or Goertzel's algorithm, these have complexity O(NM). While I did not check the math fully for correctness, the development appears correct.

Having said that, the paper completely discounts fair comparisons with the Goertzel algorithm. When M is small compared to N ($o(\log N)$), complexity of O(NM) is not much of a problem, and Goertzel's algorithm has very favorable hidden constants and is exact. If M is $O(N)$ then the proposed approach has the same complexity with the FFT, which is also exact and has favorable constants. A discussion and an experimental study of the performance of the method compared to the Goertzel algorithm or the direct implementation would demonstrate the benefits of the algorithm.

My main objection to this paper is one of scope. I do not see how the paper fits in learning conference. It would be a much better fit in a signal processing conference/journal or in a theoretical computer science one.

---

> ### Author Response · Authors · 2020-11-23
> **Response to AnonReviewer1**
>
> We would like to thank you for high quality review and constructive comments.
>
> **Comparison with Goertzel algorithm**
>
> As the reviewer mentioned, we added the discussion of PFT compared to the Goertzel algorithm both theoretically and experimentally in the second paragraph of Section 2 of the revised paper. In the discussion, we showed that PFT outperforms the Goertzel algorithm significantly, which demonstrates the benefits of our proposed algorithm.
>
> **Scope of the paper**
>
> Many papers presented in learning conferences, like ICML, CVPR, and KDD, exploit Fast Fourier transform (FFT). Our proposed approach, PFT, allows one to improve the performance (e.g., efficiency) of those existing learning approaches. Moreover, PFT will continue to support various research projects that suffer from the inefficiency of FFT. Therefore, we are confident that many ICLR audiences will be interested in PFT. The following references utilize FFT:
> - [1] Frank et al. Leveraging Frequency Analysis for Deep Fake Image Recognition, ICML 2020, https://arxiv.org/pdf/2003.08685.pdf
> - [2] Li et al. FALCON: A Fourier Transform Based Approach for Fast and Secure Convolutional Neural Network Predictions, CVPR 2020, https://openaccess.thecvf.com/content_CVPR_2020/papers/Li_FALCON_A_Fourier_Transform_Based_Approach_for_Fast_and_Secure_CVPR_2020_paper.pdf
> - [3] Yanchao Yang and Stefano Soatto. FDA: Fourier Domain Adaptation for Semantic Segmentation, CVPR 2020, https://openaccess.thecvf.com/content_CVPR_2020/papers/Yang_FDA_Fourier_Domain_Adaptation_for_Semantic_Segmentation_CVPR_2020_paper.pdf
> - [4] Chen et al. Compressing Convolutional Neural Networks in the Frequency Domain, KDD 2016, https://www.kdd.org/kdd2016/papers/files/rpp0534-chenA.pdf

---

### Comment · AnonReviewer3 · 2020-11-24
**Update**

I ave lowered the score from 6 to 5 because, in light of the comparison to the literature that the authors were not aware of earlier, they also need to compare between their algorithm and the existing algorithms experimentally.  This point is further justified by the fact that the relevance of this paper to the ICLR scope is quite weak, and mostly hinges on the fact that FFT is used in practice. Hence, much more effort must be invested in comparing between all the algorithms in the literature on actual data.

I would also recommend to compare on a deep learning task, with convolutional layers, in order to strengthen the relevance of the paper to ICLR.

---

### Decision · Program_Chairs · 2021-01-07
**Final Decision**

**Decision:**

Reject

**Comment:**

The major complaint about this paper was the lack of a proper comparison to previous work, both theoretically and empirically. Also, a study of the tradeoff between the accuracy and running time would significantly help this paper. Ultimately these were the main reasons for deciding to not accept the paper. The reviewers did think the algorithm was new and interesting, and so hopefully by addressing the complaints above, a future version of the paper could be more influential.